# NeuroLifting: Neural Inference on Markov Random Fields at Scale

## Abstract

Inference in large-scale Markov Random Fields (MRFs) remains challenging, with traditional approximate like belief propagation and exact methods such as the Toulbar2 solver often struggling to balance efficiency and solution quality at scale. This paper presents NeuroLifting, a novel approach that uses Graph Neural Networks (GNNs) to reparameterize MRF decision variables, enabling standard gradient descent optimization. By extending lifting techniques through neural networks, NeuroLifting achieves efficient, parallelizable optimization with a smooth loss landscape. Empirical results show NeuroLifting matches Toulbar2's solution quality on moderate scales while outperforming approximate methods. Notably, on large-scale MRFs, it demonstrates superior solutions compared to baselines with linear computational complexity growth, marking a significant advance in scalable MRF inference. The code of our model can be accessed at `https://anonymous.4open.science/status/NeuroLifting-5BC0`.

## 1 Introduction

Markov Random Fields stand as a fundamental computational paradigm for modeling complex dependencies among a large collection of variables, permeating a variety of domains such as computer vision (Wang et al., 2013; Su et al., 2021), natural language processing (Almutiri & Nadeem, 2022; Ammar et al., 2014; Lin et al., 2020), and network analysis (Wu et al., 2020; Yunfei Ma & Razavi, 2022). MRF's capacity to encode intricate probabilistic interactions underscores its widespread utility. However, unraveling the optimal configurations in high-dimensional settings remains a formidable task owing to the inherent computational complexity involved.

Traditional inference methodologies for MRFs bifurcate into approximate and exact strategies, each with its own set of advantages and limitations. Approximate inference techniques, such as belief propagation (Pearl, 2022; Wainwright et al., 2005) and mean field (Saito et al., 2012; Zhang, 1993) approximations, strive for computational efficiency but often at the expense of solution quality, particularly as the scale of the problem escalates. Conversely, exact inference methods, epitomized by the Toulbar2 solver (De Givry, 2023; Hurley et al., 2016), aspire to optimality but are frequently hampered by exponential time complexities that render them infeasible for large-scale MRFs.

Despite significant advances, achieving a harmonious balance between efficiency and solution quality in large-scale MRF inference has remained a long-standing unmet challenge. This paper addresses this pivotal issue through the introduction of "NeuroLifting" – a neural-network-driven paradigm that extends traditional lifting technique in the context of optimization (Albersmeyer & Diehl, 2010; Balas & Perregaard, 2002; Bauermeister et al., 2022). NeuroLifting is a novel approach that reimagines MRF inference by leveraging the potency of GNNs alongside gradient-based optimization techniques.

The core innovation of NeuroLifting lies in the reparameterization of the decision variables within MRFs utilizing a randomly initialized GNN. While some recent heuristics succeeded in utilizing GNNs for solving combinatorial problems (Cappart et al., 2023; Schuetz et al., 2022), an effective adaptation to MRF inference remains opaque. Besides, they generally lack an in-depth understanding of how GNNs facilitate downstream computation. In this paper, we for the first time bridge such practice to traditional lifting techniques, and further demonstrate that by harnessing the continuous and smooth loss landscape intrinsic to neural networks, NeuroLifting simplifies the

optimization process for large-scale MRFs, enabling enhanced parallelization and performance on GPU devices.

Empirical evaluations substantiate the efficacy of NEUROLIFTING, showcasing its ability to deliver high-quality solutions across diverse MRF datasets. Notably, it outperforms existing approximate inference strategies in terms of solution quality without sacrificing computational efficiency. When juxtaposed with exact strategies, NEUROLIFTING demonstrates comparable solution fidelity while markedly enhancing efficiency. For particularly large-scale MRF problems, encapsulating instances with over 50,000 nodes, NEUROLIFTING exhibits a linear computational complexity increase, paired with superior solution quality relative to exact methods.

In summary, the contributions of this paper are threefold. 1) **Methodical design**: we present NEUROLIFTING as an innovative and practical solution to the enduring challenge of efficient and high-quality inference in large-scale MRFs; 2) **Non-parametric lifting**: we extend the concept of lifting from traditional optimization practices into a modern neural network framework, thereby offering a fresh lens through which to tackle large-scale inference problems; 3) **Significant performance**: NEUROLIFTING achieved significant performance improvement over existing methods, showing remarkable scalability and efficiency in real-world scenarios.

## 2 RELATED WORK

**Lifting in Optimization.** Lifting techniques have significantly impacted optimization, especially for combinatorial problems and algorithm enhancement (Marchand et al., 2002). These methods transform problems into higher-dimensional spaces for better representation and solutions. Applications include mixed 0-1 integer programming as shown by Balas et al. (1993) and MIP problems with primal cutting-plane algorithms as demonstrated by Dey & Richard (2008). Integration with variable upper bound constraints has proven effective for problems like Knapsack (Shebalov et al., 2015). The techniques have expanded to robust optimization (Georghiou et al., 2020; Bertsimas et al., 2019) and shown promise when combined with Newton's method for NLPs (Albersmeyer & Diehl, 2010).

**Unsupervised GNNs for Combinatorial Optimization.** Graph Neural Networks have demonstrated their power in optimization (Yu et al., 2019; Ying et al., 2024), with recent unsupervised GNN advancements showing effectiveness in combinatorial optimization. Unsupervised GNNs can learn meaningful representations of nodes and edges without labeled data, effectively capturing combinatorial structure as shown by Peng et al. (2021). This approach has proven particularly valuable for problems like the Traveling Salesman Problem (Gaile et al., 2022; Min et al., 2023), Vehicle Routing Problem (Wu et al., 2024), and Boolean satisfiability problem (Cappart et al., 2023). Efficient solutions for Max Independent Set and Max Cut problems were also demonstrated by Schuetz et al. (2022). However, the loss functions may lack flexibility in handling higher-order relationships beyond edges.

**MRF and Inference.** The maximum a posterior (MAP) problem of MRFs is finding the best configuration that could minimize the energy function which is a NP-hard problem. Currently, the popular methods include variants of belief propagation (Weiss & Freeman, 2001; Felzenszwalb & Huttenlocher, 2004; Frey & Mackay, 2002) and tree-reweighted message passing (TRBP) (Wainwright et al., 2005; Kolmogorov, 2006) and a generalization of TRBP which is suitable for high-order MRFs(SRMP) (Kolmogorov, 2015).

Neural Networks with supervised learning remain the mainstream approach for MRF problems. GNNs were used for MRF inference by Yoon et al. (2019), outperforming LBP and TRBP but limited to 16 nodes. BP information was integrated into GNNs to formulate a neural factor graph by Garcia Satorras & Welling (2021), surpassing traditional BP on graphs under 100 nodes. GNN message passing rules were modified by Kuck et al. (2020) to align with BP properties, showing better performance than LBP on graphs up to 196 nodes. A neural version of the Max-product algorithm was proposed by Zhang et al. (2020). Variational MPNN for MAP problems on 9-13 node graphs was introduced by Cui et al. (2022). Researchers also explored CRFs. The semi-supervised method by Qu et al. (2019) used CRF to enhance GNN classification, further implemented by Tang et al. (2021). However, obtaining optimal configurations in large-scale MRFs remains challenging. Unsupervised and self-supervised learning offer alternative approaches. Learning to optimize prin-

ciples (Nair et al., 2021) were applied to MRF problems, while the Augmented Lagrangian Method was employed by Arya et al. (2024b) as loss function for self-supervised models solving CMPE problems. A self-supervised approach for binary node MMAP problems was proposed by Arya et al. (2024a). These methods require specially designed loss functions and are all limited to smaller instances.

**Our motivation**: Our research aims to develop a more comprehensive method applicable to both pairwise and high-order MRFs, capable of handling nodes with arbitrary label counts. We seek to create an approach that is not only easily implementable but also directly compatible with MRF problem frameworks. Most crucially, our method aims to maintain high performance on large-scale instances, addressing a significant gap in current methodologies.

## 3 PRELIMINARY

**Markov Random Field**. An MRF is defined over a undirected graph $\mathcal{G} = (\mathcal{V}, \mathcal{C})$, where $\mathcal{V}$ represents the index set of random variables and $\mathcal{C} \subseteq 2^{\mathcal{V}}$ is the clique set representing the (high-order) dependencies among random variables. Throughout this paper, we associate a node index $i$ with a random variable $x_i \in \mathcal{X}$, where $\mathcal{X}$ is a finite alphabet. Thus, given graph $\mathcal{G}$, the joint probability of a configuration of $X = \{x_i\}_{i \in \mathcal{V}}$ can be expressed as Eq. 1,

$$\mathbb{P}(X) = \frac{1}{Z} \exp(-E(X)) = \frac{1}{Z} \exp\left(-\sum_{i \in \mathcal{V}} \theta_i(x_i) - \sum_{C_k \in \mathcal{C}} \theta_{C_k}(\{x_l | \forall x_l \in C_k\})\right) \tag{1}$$

where $Z$ is the partition function, $\theta_i(\cdot)$ denotes the unary energy functions, $\theta_C(\cdot)$ represent the clique energy functions. In this sense, MRF provides a compact representation of probability by introducing conditional dependencies:

$$\mathbb{P}(x_i | X \setminus \{x_i\}) = \mathbb{P}(x_i | \{x_j\}) \text{ for } i, j \in C_k \text{ for } C_k \in \mathcal{C}. \tag{2}$$

In this paper, we consider the MAP estimate of Eq. 1, which requests optimizing Eq. 1 via $X^* = \min_X E(X)$. One can consult Koller & Friedman (2009) for more details.

**Graph Neural Networks**. GNNs represent a distinct class of neural network architectures specifically engineered to process graph-structured data (Kipf & Welling, 2017; Hamilton et al., 2017; Xu et al., 2019; Veličković et al., 2018). In general, when addressing a problem involving a graph $\mathcal{G} = (\mathcal{V}, \mathcal{E})$, where $\mathcal{E}$ is the edge set, GNNs utilize both the graph $\mathcal{G}$ and the initial node representations $\{h_i^{(0)} \in \mathbb{R}^d | \forall i \in \mathcal{V}\}$ as inputs, where $d$ is the dimension of initial features. Assuming the total number of GNN layers to be $K$, at the $k$-th layer the graph convolutions typically read:

$$h_i^{(k)} = \sigma\left(W_k \cdot \text{AGGREGATE}^{(k)}\left(\left\{h_j^{(k-1)} : j \in \mathcal{N}(i) \cup \{i\}\right\}\right)\right) \tag{3}$$

where $\text{AGGREGATE}^{(k)}$ is defined by the specific model, $W_k$ is a trainable weight matrix, $\mathcal{N}(i)$ is the neighborhood of node $i$, and $\sigma$ is a non-linear activation function, e.g., ReLU.

**Optimization with Lifting**. Lifting is a sophisticated technique employed in the field of optimization to address and solve complex problems by transforming them into higher-dimensional spaces (Balas, 2005; Papadimitriou & Steiglitz, 1982). By introducing auxiliary variables or constraints, lifting serves to reformulate an original optimization problem into a more tractable or elucidated form, often making the exploration of optimal solutions more accessible. In the context of MRFs, lifting can be utilized to transform inference problems into higher dimensions where certain properties or symmetries associated with specific MRF problems are more easily exploitable (Wainwright et al., 2005; Globerson & Jaakkola, 2007; Bauermeister et al., 2022). However, a principled lifting technique is still lacking for generalized MRFs.

## 4 METHODOLOGY

### 4.1 OVERVIEW

An overview of NEUROLIFTING is in Fig. 1, with an exemplary scenario involving an energy function devoid of unary terms, yet comprising three clique terms. Initially, the clique-based represen-

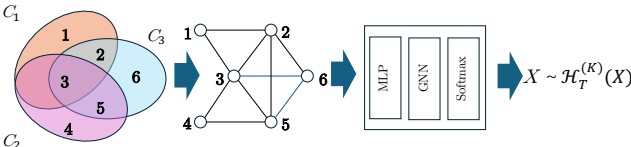

Figure 1: An overview of NEUROLIFTING. The energy function of this problem is $E(X) = \theta_{C_1}(x_1, x_2, x_3) + \theta_{C_2}(x_3, x_4, x_5) + \theta_{C_3}(x_2, x_3, x_5, x_6)$. $H_T^{(K)}$ is the output of the model after the $T$-th iteration.

tation of this function (depicted in the leftmost shaded diagram) undergoes a transformation to a graph-based perspective, which subsequently integrates into the network architecture. To address the absence of inherent node feature information in the original problem, we elevate the dimensionality of decision variables within this framework. This transformation facilitates a paradigm shift from the identification of optimal state values to the learning of optimal parameters for encoding and classification of these variables. Furthermore, we devised a novel approach to circumvent the absence of a traditional loss function, thereby extending the applicability of our framework to MRFs of arbitrary order.

## 4.2 PREPOSSESSING

We discuss several necessary preprocessing steps to adapt standard MRF to a GNN style.

**Topology construction for GNNs.** In an MRF instance, the high-order graph structure consists of nodes and cliques, diverging from typical GNNs allowing only pairwise edges (2nd-order). To facilitate the power of GNNs, we need to convert high-order graph into a pairwise one.

By the very definition of a clique, any two nodes that appear within the same clique are directly related. Thus, for any two nodes $i, j \in C_k$ in a clique $C_k$, we add a pairwise edge $(i, j)$ to its GNN-oriented graph. An example can be observed in Fig. 1. It is worth noting that an edge may appear in multiple cliques; however, we add each edge only once to the graph.

**Initial feature for GNNs.** As there is no initial features associated to MRF instances, we initialize feature vectors to GNNs *randomly* with a predefined dimension $d$. Detailed information on how we will handle these artificial features to ensure they effectively capture the underlying information of the problem will be provided in Section. 4.3.

**Vectorizing the energy function.** The transformed energy function $E(X)$ will serve as the loss function guiding the training of the neural networks. In Section. 4.4, we will detail the transformation process and discuss how to effectively utilize it. Note the values of these functions can be pre-evaluated and repeatedly used during the training process. Therefore, we employ a look-up table to memorize all function values with discrete inputs. For unary energies, we denote the vectorized unary energy of variable $x_i$ as $\phi(x_i)$, where the $n$-th element corresponds to $\theta_i(x_i = n)$. Similarly, we represent the clique energy for clique $C_k$ using the tensor $\psi(\{x_l | \forall x_l \in C_k\})$. This tensor can be derived using the same conceptual framework; for instance, the element $\psi(x_i, x_j, x_k)$ at position $(0, 2, 4)$ corresponds to the value of $\theta_{\{i,j,k\}}(x_i = 0, x_j = 2, x_k = 4)$.

**Padding node embeddings & energy terms and Masking.** GNNs typically require all node embeddings to be of the same dimension, meaning that the embeddings $h^{(K)}$ at $K$-th layer must share the same size. However, in general MRFs, the variables often exhibit different numbers of states. While traditional belief-propagation-based methods can easily manage such variability, adapting GNNs to handle these discrepancies is less straightforward.

To address this mismatch, we employ padding strategy – a common technique used to handle varying data lengths. This strategy is applied to both the node embeddings and the unary and pairwise (or clique) energies, to ensure consistent embedding dimensions. Concretely, we assign virtual states to the nodes whose state number is less than $|\mathcal{X}|$. Then, we assign energies to those padded labels with the *largest value of the original energy term*. The schematic diagram of the padding procedure could be found in Appendix A. This approach of assigning high energies to the padded labels serves to discourage the model from selecting these padded states, thereby incentivizing it to choose the original, non-padded states with lower energies. We employ a masking strategy to exclude padded

regions(using -inf as the mask), thereby ensuring these artificially added areas neither participate in the selection process nor significantly affect the loss computation.

*Remark* 4.1. Although utilizing uniform large values(e.g., padding with inf) for energy padding is theoretically viable, this approach introduces significant computational bias. Specifically, this padding methodology causes substantial deviation between the training loss and the true energy metrics, as minimal variations in padded regions disproportionately influence the loss function. Such distortion impedes effective monitoring of the training dynamics. As empirically validated in Section 5, our proposed padding scheme demonstrates superior performance by maintaining a high degree of consistency between the training loss and the true energy measurements, thereby ensuring more reliable model optimization.

### 4.3 GNNs as Non-parametric Lifting

In this section, we detail how NEUROLIFTING generates features that capture the hidden information of the given MRF and solves the original MAP problem by optimizing in a high-dimensional parameter space. As mentioned in Section 4.2, we initially generate *learnable* feature vectors randomly using an encoder that embeds all nodes, transforming the integer decision variables into $d_l$-dimension vectors $h_i^{(0)} \in \mathbb{R}^{d_l}$ for node $i$, where $d_l$ is a hyper-parameter representing the dimension after lifting.

The intuition for utilizing GNNs in the implementation of lifting techniques is inspired by LBP. When applying LBP for inference on MRFs, the incoming message $M_{ji}$ to node $i$ from node $j$ is propagated along the edges connecting them. Node $i$ can then update its marginal distributions according to the formula in Eq. 4 where $exp(-\phi(x_i))$ is the unary potential function.

$$p^{\textbf{posterior}}(x_i|X\backslash\{x_i\}) = \exp(-\phi(x_i)) \prod_{(i,j)\in\mathcal{E}} \sum_{x_j} M_{ji} \qquad (4)$$

Importantly, the incoming messages are not limited to information solely about the directly connected nodes; they also encompass information from sub-graphs that node $i$ cannot access directly without assistance from its neighbors. This allows a more comprehensive aggregation of information, enabling node $i$ to merge these incoming messages with its existing information. This process of message aggregation bears resemblance to the message-passing procedure used in GNNs, where nodes iteratively update their states based on the information received from their neighbors.

Graph convolutions should intuitively treat adjacent nodes equally, consistent with the principle in MRFs, where the information collected from neighbors is processed equally. Typical GNNs are summarized in Table 8 from Appendix 8, where $\deg(i)$ is the degree of node $i$, $\alpha_{i,j}$ is the attention coefficients, and $|\mathcal{N}(i)|$ is the neighborhood size of node $i$. According to the influence of neighbors, they can be classified into three categories: 1) neighborhood aggregation with normalizations (e.g., GCN (Kipf & Welling, 2017) normalize the influence by node degrees), 2) neighborhood aggregation with directional biases (e.g., GAT (Veličković et al., 2018) learn to select the important neighbors via an attention mechanism), and 3) neighborhood aggregation without bias (e.g., Graph-SAGE (Hamilton et al., 2017) directly aggregate neighborhood messages with the same weight). Therefore, we select the aggregator in GraphSAGE as our backbone for graph convolutions. The performance of these GNN backbones on our MRF datasets is shown in Fig. 5 in Appendix G.

Another primary characteristic of MRFs is its ability to facilitate information propagation across the graph through local connections. This means that even though the interactions are defined locally between neighboring nodes, the influence of a node can extend far beyond its immediate vicinity. As a result, MRFs can effectively capture global structure and dependencies within the data. We thus use Jumping Knowledge (Xu et al., 2018) to leverage different neighborhood ranges. By doing so, features representing local properties can utilize information from nearby neighbors, while those indicating global states may benefit from features derived from higher layers.

At each round of iterations, we optimize both the GNN parameters and those of the encoder. At the start of the next iteration, we obtain a new set of feature vectors, $\mathcal{H}_t^{(0)} = \{h_{i,t}^{(0)} \in \mathbb{R}^{d_l}|\forall i \in \mathcal{V}\}$, where $t$ indicates the $t$-th iteration. This process enables us to accurately approximate the latent features of the nodes in a higher-dimensional space.

### 4.4 ENERGY MINIMIZATION WITH GNN

As indicated by Eq. 1, the energy function can serve as the loss function to guide network training since minimizing this energy function aligns with our primary objective. Typically, the energy function for a new problem instance takes the form of a look-up table, rendering the computation process non-differentiable. To facilitate effective training in a fully unsupervised setting, it is crucial to transform this computation into a differentiable loss aligning with the original energy function. The initial step involves transforming the decision variable from $x_i \in \{1, ..., s_i\}$, where $s_i$ is the number of states of variable $x_i$, to $v_i \in \{0, 1\}^{s_i}$. At any given time, exactly one element of the vector $v_i$ can be one, while all other elements must be zero; the position of the 1 indicates the current state of the variable $x_i$. Define $V_k = \otimes_{i \in C_k} v_i$, where $\otimes$ is the tensor product. The corresponding energy function would be Eq. 5. Subsequently, we relax the vector $v_i$ to $p_i(\theta) \in [0, 1]^{s_i}$, where $p_i(\theta)$ represents the output of our network and $\theta$ denotes the network parameters. This output can be interpreted as the probabilities of each state that the variable $x_i$ might assume.

$$E(\{v_i | i \in \mathcal{V}\}) = \underbrace{\sum_{i \in \mathcal{V}} \langle v_i(\theta), \phi(x_i) \rangle}_{\text{Unary Term}} + \underbrace{\sum_{C_k \in \mathcal{C}} \langle \psi(C_K), V_k \rangle}_{\text{Clique Term}} \tag{5}$$

$$L(\theta) = \underbrace{\sum_{i \in \mathcal{V}} \langle p_i(\theta), \phi(x_i) \rangle}_{\text{Unary Term}} + \underbrace{\sum_{C_k \in \mathcal{C}} \langle \psi(C_K), P_k \rangle}_{\text{Clique Term}} \tag{6}$$

$$L_{\text{Cross Entropy}} = -\sum_i Q_i log(P_i) \tag{7}$$

where $\langle \cdot, \cdot \rangle$ refers to the tensor inner product. The applied loss function is defined in Eq. 6, here $P_k = \otimes_{i \in C_k} p_i$. The rationale behind our loss function closely resembles that of the cross-entropy loss function commonly used in supervised learning. Let $P_i$ represent the true distribution and $Q_i$ denote the predicted distribution of the node $i$. A lower value of cross-entropy Eq. 7 indicates greater similarity between these two distributions. However, our approach differs in that we are not seeking a predicted distribution that closely approximates the true distribution. Instead, for each variable, we aim to obtain a probability distribution that is highly concentrated, with the concentrated points corresponding to the states that minimize the overall energy.

Once the network outputs are available, we can easily determine the assignments by *rounding* the probabilities $p(\theta)$ to obtain binary vectors $v$. Using these rounded results, the actual energy can be calculated using Eq. 5. It is observed that after the network converges, the discrepancy between $L(\theta)$ and $E(\{v_i | i \in \mathcal{V}\})$ is minor and we won't see any multi-assignment issue in decision variables. We choose Adam (Kingma & Ba, 2015) as the optimizer, and employ simulated annealing during the training process, allowing for better exploring the loss landscape to prevent sub-optima.

### 4.5 ANALYSIS AND DISCUSSION

**Relation to lifting.** In this innovative framework of using GNNs for inference on MRFs, a natural and sophisticated parallel emerges with the classical concept of lifting in optimization (Balas et al., 1993). By mapping each unary term of an MRF to a node within a GNN and translating clique terms into densely connected subgraphs, the traditional MRF energy minimization transforms into optimizing a multi-layer GNN with extra dimensionality. This procedure aligns with the lifting technique where the problem space is expanded to facilitate more efficient computation. Akin to the principle of standard lifting to ease optimization, the GNN-based reparameterization can leverage the gradient descent optimization paradigm inherent in the smooth neural network landscape (Dauphin et al., 2014; Choromanska et al., 2015), ensuring efficient computation and convergence. Therefore, while offering an enhanced approach to inference, the GNN reparameterization mirrors the core principles of lifting by transforming and extending the solution space into a computation-friendly one to achieve computational efficacy and scalability. More empirical evidence is in Section 5.4.

**Complexity analysis.** The primary computations in this model arise from both the loss calculation and the operations within the GNN. For the loss function, let $c_{max}$ denote the maximum clique size. The time complexity for the loss calculation is given by $O(|\mathcal{V}||\mathcal{X}| + c_{max}|\mathcal{C}||\mathcal{X}|)$. For the GNN component, let $\mathcal{N}_v$ denote the average number of neighbors per node in the graph. The time complexity for neighbor aggregation in each layer is $O(\mathcal{N}_v|\mathcal{V}|)$, and merging the results for all nodes

requires $O(|\mathcal{V}|d)$ where $d$ is the feature dimension. Thus, for a $K$-layer GraphSAGE model with the custom loss function, the overall time complexity can be expressed as $O(|\mathcal{X}|(|\mathcal{V}| + c_{max}|\mathcal{C}|) + K|\mathcal{V}|(\mathcal{N}_v + d))$. This analysis highlights the efficiency of the framework in managing large-scale graphs by leveraging neighborhood sampling and aggregation techniques. The derived complexity indicates that the model scales linearly with respect to the number of nodes, the number of layers, and the dimensionality of the feature vectors, making it well-suited for large-scale instances.

## 5 EXPERIMENT

**Evaluation metric.** For all instances used in the experiments, we utilize the final value of the overall energy function $E(X)$ as defined in Eq. 1. Without loss of generality, all problems are formulated as minimization problems.

**Baselines.** We compare our approach against several well-established baselines: Loopy Belief Propagation (LBP), Tree-reweighted Belief Propagation (TRBP) (Wainwright et al., 2005), and Toulbar2 (De Givry, 2023). LBP is a widely used approximate inference algorithm that iteratively passes messages between nodes. TRBP improves upon LBP by introducing tree-based reweighing to achieve better approximations, particularly in complex graph structures. Toulbar2 is an exact optimization tool based on constraint programming and branch-and-bound methods Notably, Toulbar2 is the winner on **all** MPE and MMAP task categories of UAI 2022 Inference Competition [1]. These baselines allow us to evaluate the performance of our proposed solution under fair settings.Note that comparisons with LBP and TRBP are omitted for high-order cases, as these methods are limited to simple scenarios on this kind of problems. We will use SRMP (Kolmogorov, 2015) on the high-order cases instead.

**MRF format and transformation.** The MRF data files are in UAI format and we interpret the data files in the same way as Toulbar2. Detailed information about unary and clique terms will be treated as unnormalized (joint) distributions, and the energies are calculated as $\theta_i(x_i = a) = -log(P(x_i = a))$, where $P(x_i = a)$ represents the probability provided by the data file. Note that we use the unnormalized values during the transformation process. The transformation for the clique energy terms will follow the same procedure. More details are in Appendix I.

### 5.1 SYNTHETIC PROBLEMS

We first conduct experiments on synthetic problems generated randomly based on Erdős–Rényi graphs (Erdös & Rényi, 1959). The experiments are divided into pairwise cases and higher-order cases. We will compare the performance of NEUROLIFTING with LBP, TRBP, and Toulbar2 on pairwise MRFs. For the higher-order MRF cases, we will compare NEUROLIFTING exclusively with Toulbar2, as LBP and TRBP are not well-suited for handling the complexities inherent in high-order MRFs. The raw probabilities (energies) on the edges/cliques are randomly generated using the Potts function (Eq. 8), representing two typical types found in the UAI 2022 dataset. The parameters $\alpha$ and $\beta$ serve as constant penalty terms and $\mathbb{I}$ is the indicator function.

$$\theta_{ij} = \alpha\mathbb{I}(x_i = x_j) + \beta \tag{8}$$

For all the random cases, all the probabilities values of the unary terms and pairwise (clique) terms are generated randomly range from 0.2 to 3.0. For the Potts models, $\alpha, \beta \in [0.00001, 1000]$. Each random node can select from 2 to 6 possible discrete labels, and the values of the unary terms are also generated randomly, ranging from 0.2 to 3.0. LBP and TRBP are allowed up to 60 iterations, with a damping factor 0.1 to mitigate potential oscillations. Toulbar2 operates in the default mode with time limit 18000s. We employ a 5-layer GNN to model all instances and $d_l = 1024$. The learning rate is set to $1e^{-4}$, and the model is trained for up to 150 iterations for each instance, utilizing a simple early stopping rule with an absolute tolerance of $1e^{-4}$ and a patience of 10. We will give 5 trails to NEUROLIFTING to eliminate randomness. The data generation method and the parameter settings are the same for both pairwise cases and high order cases.

**Pairwise instances.** The inference results on pairwise cases are summarized in Table. 1. Due to the page limits here we only show the best results. The full table of the results with more statistics is shown in Table 4 in Appendix C. Prefix "P_potts_" and "P_random_" indicate instances generated

---

[1]https://www.auai.org/uai2022/uai2022_competition

with Potts energy and random energy, respectively. It is evident that as the problem size scales up, NEUROLIFTING outperforms the baseline approaches; meanwhile, it also achieves comparable solution quality even when the problem sizes are small. This trend is consistent across both energy models.

Table 1: Results on pairwise synthetic instances. Numbers are the energy values. Best in bold.

| Graph | #Nodes/#cliques | LBP | TRBP | Toulbar2 | NEUROLIFTING | Graph | #Nodes/#cliques | LBP | TRBP | Toulbar2 | NEUROLIFTING |
|---|---|---|---|---|---|---|---|---|---|---|---|
| P_potts_1 | 1k/7591 | -22215.700 | -21365.800 | **-22646.529** | -21451.025 | P_random_1 | 1k/7540 | **-4901.100** | -4505.020 | -4900.759 | -4564.763 |
| P_potts_2 | 5k/37439 | **-111319.000** | -105848.000 | -110022.248 | -105952.531 | P_random_2 | 5k/37488 | -24059.900 | -22934.000 | **-24139.194** | -21834.693 |
| P_potts_3 | 10k/75098 | **-221567.000** | -210570.000 | -218311.424 | -209925.269 | P_random_3 | 10k/74518 | -47873.200 | -47002.000 | **-48107.172** | -42120.325 |
| P_potts_4 | 50k/248695 | 12411.200 | 13454.600 | 12955.129 | **11679.429** | P_random_4 | 50k/249554 | 12881.500 | 14342.300 | 12233.890 | **11769.934** |
| P_potts_5 | 50k/249624 | 25668.500 | 35389.000 | 12468.172 | **11466.507** | P_random_5 | 50k/249374 | 12478.300 | 13337.000 | 12835.994 | **11750.969** |
| P_potts_6 | 50k/300181 | 17609.800 | 17362.600 | 17635.791 | **16756.999** | P_random_6 | 50k/299601 | 16723.600 | 16754.500 | 18031.964 | **16700.674** |
| P_potts_7 | 50k/299735 | **16962.500** | **16962.500** | 19532.817 | 17002.578 | P_random_7 | 50k/299538 | **16689.200** | 16701.600 | 18179.548 | **16689.252** |
| P_potts_8 | 50k/374169 | **24552.400** | 24596.600 | 25446.235 | **24552.413** | P_random_8 | 50k/374203 | **24556.000** | **24556.000** | 25549.594 | **24555.995** |
| P_potts_9 | 50k/375603 | 25099.800 | 25095.600 | 25502.495 | **25050.522** | P_random_9 | 50k/374959 | **24635.600** | 24689.500 | 25908.500 | 24640.039 |

**Higher-order instances.** The inference results on high order cases are summarized in Table. 2. The "H" in the prefix stands for High-order and all the instances are generated using Potts model. The number of cliques in the table encompasses both the cliques themselves and the edges connecting them. The relationships between nodes are based on either pairwise interactions or clique relationships. The results indicate that NEUROLIFTING outperforms Toulbar2 and SRMP, demonstrating its ability to effectively handle complicate high-order MRFs. This performance highlights the robustness and effectiveness of NEUROLIFTING across different graph structures.

## 5.2 UAI 2022 INFERENCE COMPETITION DATASETS

We then evaluate our algorithm using instances from the UAI 2022 Inference Competition datasets, including both pairwise cases and high-order cases. The time settings will align with those established in the UAI 2022 Inference Competition, specifically 1200 seconds and 3600 seconds. LBP and TRBP algorithms are set to run for 30 iterations with a damping factor of 0.1, and the time limit for Toulbar2 is configured to 1200 seconds, which is generally sufficient for convergence. For NEUROLIFTING, we utilize an 8-layer GNN to model all instances, with the model trained for up to 100 iterations for each instance; other settings remain consistent with those used in the synthetic problems. We also experimented with lifting dimensions of 64, 512, 1024, 4096, and 8192.

**Pairwise cases.** We evaluated pairwise cases from the UAI MPE dataset, with results in Appendix D. Table 5 shows NEUROLIFTING achieves solutions comparable to LBP and TRBP on trivial problems where Toulbar2 finds optimal solutions. On more challenging problems, while not surpassing Toulbar2, NEUROLIFTING outperforms both LBP and TRBP, indicating better performance on real-world datasets than artificial instances. Complete results with varying lifting dimensions appear in Appendix D.

**High-order cases.** For the high-order cases, we select a subset that has relatively large sizes. The results are presented in Table 6 in Appendix D. The performance of NEUROLIFTING aligns with the results obtained from synthetic instances, demonstrating superior efficacy on larger problems while consistently outperforming Toulbar2 in dense cases.

## 5.3 PHYSICAL CELL IDENTITY

Physical Cell Identity (PCI) uniquely identifies cells in LTE and 5G networks, distinguishing between neighboring cells. We transform PCI instances into pairwise MRFs to enable evaluation across all baselines. Transformation details are provided in Appendix J. We evaluated using internal real-world PCI data and synthetic datasets. LBP, TRBP, and NEUROLIFTING configurations match

Table 2: Results on the synthetic **high order** MRFs. Numbers correspond to the energy values. Best in bold. "NA" denotes that no solution was found within the specified time limits. Best in bold.

| Graph | #Nodes/#cliques | Toulbar2 | SRMP | NEUROLIFTING | |
|---|---|---|---|---|---|
| | | | | Energy | Loss |
| H_Instances_1 | 500/41253 | NA | -5785.093 | **-7866.214 ± 389.207** | -7859.68 ± 393.719 |
| H_Instances_2 | 500/57934 | NA | -18504.788 | **-20260.289 ± 143.276** | -20286.571 ± 143.624 |
| H_Instances_3 | 1000/36993 | NA | -5903.131 | **-7232.648 ± 337.393** | -7229.483 ± 336.218 |

Table 3: Results on the PCI instances. Numbers are the obtained energy values. Best in bold.

| Graph | #Nodes/#Cdges | LBP | TRBP | Toulbar2 | NEUROLIFTING | |
|---|---|---|---|---|---|---|
| | | | | | Energy | Loss |
| PCI_1 | 30/165 | 20.344 | 20.455 | **18.134** | $18.372 \pm 0.161$ | $18.373 \pm 0.160$ |
| PCI_2 | 40/311 | **98.364** | 98.762 | **98.364** | $98.555 \pm 0.109$ | $98.555 \pm 0.109$ |
| PCI_3 | 80/1522 | **1003.640** | **1003.640** | **1003.640** | $1003.640 \pm 0.0$ | $1003.639 \pm 0.0$ |
| PCI_4 | 286/10714 | 585.977 | 585.977 | 426.806 | $\mathbf{410.945 \pm 2.009}$ | $410.996 \pm 2.014$ |
| PCI_5 | 929/29009 | 1591.590 | 1591.590 | 1118.097 | $\mathbf{1074.617 \pm 5.501}$ | $1074.676 \pm 5.503$ |
| PCI_synthetic_1 | 280/9678 | 564198.000 | 568082.000 | 522857.923 | $\mathbf{496015.5 \pm 6307.363}$ | $496013.662 \pm 6297.169$ |
| PCI_synthetic_2 | 526/34500 | 2.092e+06 | 2.084e+06 | 2.064e+06 | $\mathbf{1.923e{+}06 \pm 9977.015}$ | $1.923e{+}06 \pm 10007.739$ |
| PCI_synthetic_3 | 1000/49950 | 2.932e+06 | 2.908e+06 | 2.856e+06 | $\mathbf{2.665e{+}06 \pm 4555.868}$ | $2.664e{+}06 \pm 4468.965$ |
| PCI_synthetic_4 | 1500/78770 | 4.568e+06 | 4.532e+06 | 4.534e+06 | $\mathbf{4.215e{+}06 \pm 13500.602}$ | $4.214e{+}06 \pm 13252.456$ |
| PCI_synthetic_5 | 2000/120024 | 6.807e+06 | 6.904e+06 | 7.023e+06 | $\mathbf{6.542e{+}06 \pm 19789.758}$ | $6.540e{+}06 \pm 19782.638$ |

Section 5.1, but with 100 iterations, while Toulbar2 had a 3600-second limit with default parameters. Table 3 shows results for five real-world cases from a Chinese city and five synthetic instances. Toulbar2 solves smaller problems exactly but struggles with larger scales. Similarly, LBP and TRBP face convergence issues on complex problems. NEUROLIFTING demonstrates strong generalization across all scales, achieving notable performance even on large instances.

## 5.4 ANALYSIS AND ABLATION STUDY

**Efficiency Analysis.** Following UAI protocol, we compared NEUROLIFTING against Toulbar2 over 1200 seconds, with metrics at 200-second intervals (due to Toulbar2's logging limitations, there is no time information on complex problems). Results in Table 9 (Appendix G) show Toulbar2 performs better on simpler instances, solving the first three within 200 seconds. However, on complex problems, Toulbar2 fails to terminate within 1200 seconds with minimal quality improvement. NEUROLIFTING maintains efficiency on larger instances, consistently outperforming Toulbar2 across all time intervals with superior solutions.

**Choice of GNN backbones.** We evaluated GNN backbones from Section 4.3 across UAI 2022 pairwise cases, private PCI instances, and synthetic datasets (1000 nodes, average degree 4/8), testing both random energy configurations and Potts models. Fig. 5 in Appendix G demonstrates Graph-SAGE's consistent superiority in both results quality and convergence speed across all datasets.

**Choice of Optimizer.** Optimizer selection, discussed in Section 4.4, is based on problem structure analysis and empirical testing. We evaluated SGD, RMSprop, and Adam on UAI 2022 pairwise cases using learning rate $10^{-4}$, 1024-dimensional embedded features, and 8-layer networks across all tests. Results in Fig. 6 (Appendix G) show Adam's superior convergence speed and stability compared to RMSprop and SGD.

**Loss Landscape Visualization.** We visualize loss landscapes using the tool from Li et al. (2018), with detailed settings in Appendix H. Fig. 4 in Appendix G shows landscape evolution for networks of depths $K \in \{1, 2, 5, 8\}$, while Fig. 7 displays converged loss trends. We observe that much of the loss function remains flat, with decreases possible only in limited parameter space regions. Deeper lifted models effectively expand these regions, enabling better solution convergence and demonstrating enhanced optimization landscape navigation capacity.

## 6 CONCLUSION

In this paper, we introduced NEUROLIFTING and its application to solving MAP problems for MRFs. Our experiments showed that NEUROLIFTING effectively handles MRFs of varying orders and energy functions, achieving performance on par with established benchmarks, as verified on the UAI 2022 inference competition dataset. Notably, NEUROLIFTING excels with large and dense MRFs, outperforming traditional methods and competing approaches on both synthetic large instances and real-world PCI instances.This method, which utilizes Neural Networks for lifting, has proven successful and could potentially be extended to other optimization problems with similar modeling frameworks.

## 7 REPRODUCIBILITY STATEMENT

To ensure full reproducibility of our research, we provide comprehensive resources and documentation. The complete source code for our implementation is available at the link provided in the abstract, along with detailed instructions for execution in the repository. All datasets used in our experiments are clearly identified in Section 5, with sources that are publicly accessible. We provide detailed configuration parameters for both our proposed algorithm and all baseline methods to enable precise replication. Appendix I includes instructions for parsing .uai files, following the standard format for MRF instance representation, while Appendix J documents our methodology for transforming PCI instances from MIP format into MRF representations. These materials collectively enable complete reproduction of our experimental results and facilitate further research in this direction.

## 8 ETHICS STATEMENT

During the preparation and submission of this paper, we have strictly adhered to the Code of Ethics in scientific research. We ensured proper citation of all relevant work, maintained integrity in our experimental procedures, reported results accurately without manipulation, and respected confidentiality of data sources where applicable. All authors have contributed substantially to this work and approved the final manuscript, with no conflicts of interest undisclosed.

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

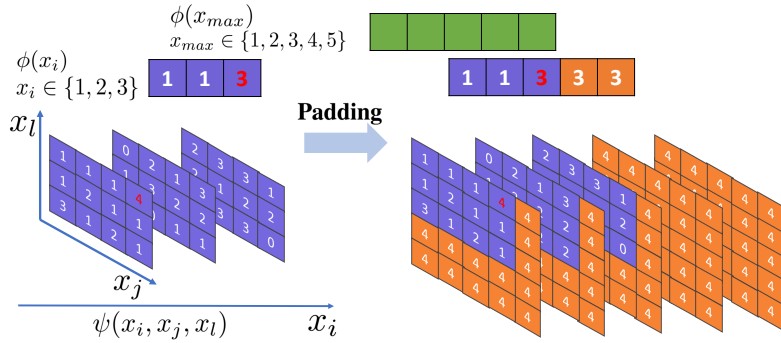

Figure 2: This illustrates the padding procedure for unary loss terms $\phi(x)$ and clique loss terms $\psi(x_i, x_j, x_k)$, with $|\mathcal{X}| = 5$. $x_{max}$ denotes the variable that has the maximum value range. The elements shown in purple represent the energy values in the original $\phi$ and $\psi$. After padding, the dimension of vector $\phi$, as well as each dimension of the energy tensor $\psi(x_i, x_j, x_k)$, will be 5. The padded portion is indicated in orange, with values either $\max(\phi)$ or $\max(\psi)$.

## A    PADDING PROCEDURE

The schematic diagram of the padding procedure is in Fig. 2. In this example, we consider the case where $|\mathcal{X}| = 5$. We start with the unary energy vector for $x_i$ denoted as $\phi(x_i) = \{1, 1, 3\}$, which has three states. Before padding, the highest value in this vector is 3, highlighted in red, and this value will be used for padding. The padded vector is shown on the right-hand side of the figure, with the padded portion indicated in orange. For the clique terms, we will apply padding similarly to the unary terms. The original energy matrix for the clique involving nodes $i, j, l$ has a dimension of $3 \times 3 \times 4$. Given that $|\mathcal{X}| = 5$, we need to pad the matrix so that $\psi(x_i, x_j, x_l) \in R^{5 \times 5 \times 5}$. In this case, the largest value in the original energy matrix is 4. As depicted in the figure, all padded values in the orange area are filled with 4.

## B    LIMITATIONS

Our proposed GNN-based approach, while effective for complex MRF problems, presents several limitations worth acknowledging. The method's computational overhead makes it less efficient for small instances where traditional algorithms may perform adequately without the preprocessing and inference costs of neural networks. Additionally, memory requirements for maintaining graph structures during message passing can become prohibitive for extremely large MRFs. Future work should focus on fully fully leveraging the potential of our method and balancing the trade-off between powerful representations and computational efficiency.

## C    FULL RESULTS ON PAIRWISE SYNTHETIC INSTANCES

In this section we show the full statistics about the 5 trials we have on the pairwise synthetic instances we have with different sizes and different energy formulations. All the result are shown in Table. 4.

## D    RESULTS OF UAI INFERENCE COMPETITION 2022 DATASET

Table. 5 and Table. 6 shows the final results of our NEUROLIFTING and the baselines on pairwise cases and high-order cases from the UAI Inference Competition 2022 separately. In Table. 7, we present the inference results of NEUROLIFTING using various dimensions of feature embeddings applied to the pairwise cases. The results indicate that the dimensionality of the feature embeddings is indeed a factor that influences model performance. However, in most cases, a moderate dimension is sufficient to achieve high-quality results. This suggests that while increasing dimensionality

Table 4: Results on ER graphs with state numbers range from 2 to 6. Numbers out of the bracket correspond to the obtained energy values, the number in the brackets is the final loss given by the loss function.

| Graph | #Nodes/#Edges | LBP | TRBP | Toulbar2 | NEUROLIFTING | |
|---|---|---|---|---|---|---|
| | | | | | Energy | Loss |
| P_potts_1 | 1k/7591 | -22215.700 | -21365.800 | -22646.529 | -21791.868 ± 218.106 | -21799.268 ± 216.075 |
| P_potts_2 | 5k/37439 | -111319.000 | -105848.000 | -110022.248 | -105762.092 ± 434.674 | -106016.855 ± 168.861 |
| P_potts_3 | 10k/75098 | -221567.000 | -210570.000 | -218311.424 | -211406.914 ± 1489.099 | -210182.681 ± 275.164 |
| P_potts_4 | 50k/248695 | 12411.200 | 13454.600 | 12955.129 | 12219.817 ± 538.969 | 11811.682 ± 123.877 |
| P_potts_5 | 50k/249624 | 25668.500 | 35389.000 | 12468.172 | 12010.036 ± 266.610 | 11673.708 ± 146.513 |
| P_potts_6 | 50k/300181 | 17609.800 | 17362.600 | 17635.791 | 18399.913 ± 1475.526 | 16988.347 ± 163.587 |
| P_potts_7 | 50k/299735 | 16962.500 | 16962.500 | 19532.817 | 17480.701 ± 212.084 | 17265.434 ± 140.904 |
| P_potts_8 | 50k/374169 | 24552.400 | 24596.800 | 25446.235 | 26115.840 ± 1677.627 | 24668.087 ± 163.587 |
| P_potts_9 | 50k/375603 | 25099.800 | 25095.600 | 25502.495 | 26348.525 ± 1095.319 | 25189.789 ± 132.693 |
| P_random_1 | 1k/7540 | -4901.100 | -4505.020 | -4900.759 | -4570.079 ± 31.228 | -4574.664 ± 31.411 |
| P_random_2 | 5k/37488 | -24059.900 | -22934.000 | -24139.194 | -21774.416 ± 52.910 | -21798.702 ± 32.389 |
| P_random_3 | 10k/74518 | -47873.200 | -47002.000 | -48107.172 | -41953.991 ± 237.577 | -41972.379 ± 216.574 |
| P_random_4 | 50k/249554 | 12881.500 | 14342.300 | 12233.890 | 12552.252 ± 30.311 | 11983.388± 213.454 |
| P_random_5 | 50k/249374 | 12478.300 | 13337.000 | 12835.994 | 12308.580 ± 14.045 | 11945.450 ± 194.481 |
| P_random_6 | 50k/299601 | 16723.600 | 16754.500 | 18031.964 | 17705.219 ± 435.560 | 17207.997 ± 405.217 |
| P_random_7 | 50k/299538 | 16689.200 | 16701.600 | 18179.548 | 18343.026 ± 1448.821 | 16971.435 ± 209.021 |
| P_random_8 | 50k/374203 | **24556.000** | **24556.000** | 25549.594 | 25949.446 ± 995.956 | 24787.343 ± 163.587 |
| P_random_9 | 50k/374959 | **24635.600** | 24689.500 | 25908.500 | 25871.264 ± 1087.915 | 24811.354 ± 171.315 |

may provide some advantages, the decision should be made by considering both performance and computational efficiency.

Table 5: Results on the UAI inference competition 2022. Numbers correspond to the obtained energy values. Best in bold."opt" denotes it is the optimal solution.

| Graph | #Nodes/#Edges | LBP | TRBP | Toulbar2 | NEUROLIFTING | |
|---|---|---|---|---|---|---|
| | | | | | Energy | Loss |
| ProteinFolding_11 | 400/7160 | -3106.080 | -3079.030 | -4461.047 | -3976.908 ± 52.047 | -4018.784 ± 36.491 |
| ProteinFolding_12 | 250/1848 | 3570.210 | 3604.240 | 3562.387(opt) | 16137.682 ± 16.020 | 16090.801 ± 22.869 |
| Grids_19 | 1600/3200 | -2250.440 | -2103.610 | -2643.107 | -2400.251 ± 20.061 | -2398.078 ± 16.010 |
| Grids_21 | 1600/3200 | -13119.300 | -12523.300 | -18895.393 | -16592.926 ± 94.368 | -16605.564 ± 113.096 |
| Grids_24 | 1600/3120 | -13210.400 | -13260.900 | -18274.302 | 16323.767 ± 171.950 | -16222.104 ± 222.593 |
| Grids_25 | 1600/3120 | -2170.890 | -2171.050 | -2620.268 | -2361.900 ± 10.231 | -2361.055 ± 12.678 |
| Grids_26 | 400/800 | -2063.350 | -1903.910 | -3010.719 | -2595.041 ± 43.306 | -2577.378 ± 39.370 |
| Grids_27 | 1600/3120 | -9024.640 | -9019.470 | -12284.284 | -10898.595 ± 160.435 | -10771.257 ± 170.329 |
| Grids_30 | 400/760 | -2142.890 | -2154.910 | -2984.248 | -2651.035 ± 35.508 | -2676.246 ± 19.886 |
| Segmentation_11 | 228/624 | 329.950 | 339.762 | 312.760 (opt) | 432.291 ± 34.208 | 391.971 ± 40.099 |
| Segmentation_12 | 231/625 | 75.867 | 77.898 | 51.151 (opt) | 90.248 ± 22.655 | 105.639 ± 21.165 |
| Segmentation_13 | 225/607 | 75.299 | 88.554 | 49.859 (opt) | 80.156 ± 6.462 | 78.685 ± 18.546 |
| Segmentation_14 | 231/632 | 95.619 | 98.691 | 92.334 (opt) | 102.263 ± 7.169 | 101.268 ± 5.467 |
| Segmentation_15 | 229/622 | 412.990 | 418.853 | 380.393 (opt) | 417.276 ± 22.357 | 408.214 ± 27.037 |
| Segmentation_16 | 228/610 | 100.853 | 101.670 | 95.000 (opt) | 102.687 ± 4.571 | 101.687 ± 7.358 |
| Segmentation_17 | 225/612 | 421.888 | 432.012 | 407.065 (opt) | 445.843 ± 24.459 | 478.881 ± 32.824 |
| Segmentation_18 | 235/647 | 100.389 | 98.411 | 82.669 (opt) | 104.721 ± 6.489 | 96.315 ± 6.124 |
| Segmentation_19 | 228/624 | 86.589 | 86.692 | 58.704 (opt) | 96.173 ± 4.731 | 84.882 ± 10.063 |
| Segmentation_20 | 232/635 | 289.435 | 291.527 | 262.216 (opt) | 335.245 ± 36.163 | 315.482 ± 24.268 |

Table 6: Results on high-order cases of the UAI inference competition 2022. Numbers correspond to the obtained energy values. Best in bold.

| Graph | #Nodes/#Cdges | Toulbar2 (1200s) | Toulbar2 (3600s) | NEUROLIFTING | |
|---|---|---|---|---|---|
| | | | | Energy | Loss |
| Maxsat_gss-25-s100 | 31931/96111 | **-145969.060** | -145969.060 | -139904.266 ± 1717.483 | -139914.341 ± 1710.139 |
| BN-nd-250-5-10 | 250/250 | 155.129 | **154.610** | 189.395 ± 5.424 | 187.729 ± 4.966 |
| Maxsat_mod4block_2vars_10gates_u2_autoenc | 479/123509 | -186103.111 | -186103.111 | 146166.620 ± 41250.035 | -146166.797 ± 41249.859 |
| Maxsat_mod2c-rand3bip-sat-240-3.shuffled-as.sat05-2520 | 339/2416 | -3734.627 | **-3737.076** | -3511.994 ± 220.654 | -3511.822 ± 220.471 |
| Maxsat_mod2c-rand3bip-sat-250-3.shuffled-as.sat05-2535 | 352/2492 | **-3863.259** | -3863.259 | -3686.567 ± 166.998 | -3686.085 ± 166.499 |

Table 7: Full results on the UAI inference competition 2022 of NEUROLIFTING with different feature dimensions. Numbers correspond to the obtained energy values.

| Graph | #Nodes/#Edges | dim=64 | dim=512 | dim=1024 | dim=4096 | dim=8192 |
|---|---|---|---|---|---|---|
| ProteinFolding_11 | 400/7160 | -3892.949 | -3886.701 | -3946.168 | 4065.294 | -4003.323 |
| ProteinFolding_12 | 250/1848 | 16064.795 | 16068.406 | 16051.798 | 16088.073 | 16071.324 |
| Grids_19 | 1600/3200 | -2355.159 | -2404.975 | -2337.281 | -2341.2746 | -2373.618 |
| Grids_21 | 1600/3200 | -16478.466 | -16169.0320 | -16446.410 | -16209.017 | -16278.668 |
| Grids_24 | 1600/3120 | -16008.008 | -15900.249 | -15841.799 | - 15608.162 | -15948.219 |
| Grids_25 | 1600/3120 | -2343.547 | -2353.223 | -2319.899 | -2306.686 | -2288.182 |
| Grids_26 | 400/800 | -2532.837 | -2608.395 | -2553.781 | -2559.572 | -2535.464 |
| Grids_27 | 1600/3120 | -10748.024 | -10704.057 | -10514.857 | -10389.031 | -10665.737 |
| Grids_30 | 400/760 | -2563.274 | -2631.862 | -2640.044 | -2691.091 | -2649.462 |
| Segmentation_11 | 228/624 | 330.541 | 349.906 | 334.882 | 356.895 | 337.312 |
| Segmentation_12 | 231/625 | 74.705 | 74.029 | 155.062 | 79.151 | 105.801 |
| Segmentation_13 | 225/607 | 67.371 | 86.064 | 69.430 | 72.394 | 112.516 |
| Segmentation_14 | 231/632 | 94.192 | 96.501 | 100.582 | 104.091 | 96.572 |
| Segmentation_15 | 229/622 | 388.223 | 386.701 | 397.246 | 407.731 | 390.641 |
| Segmentation_16 | 228/610 | 99.086 | 99.690 | 111.121 | 98.209 | 108.360 |
| Segmentation_17 | 225/612 | 424.686 | 426.130 | 425.192 | 425.240 | 427.810 |
| Segmentation_18 | 235/647 | 89.905 | 101.307 | 94.224 | 88.854 | 88.809 |
| Segmentation_19 | 228/624 | 76.244 | 78.337 | 74.284 | 69.116 | 70.770 |
| Segmentation_20 | 232/635 | 298.802 | 301.802 | 302.673 | 304.457 | 312.970 |

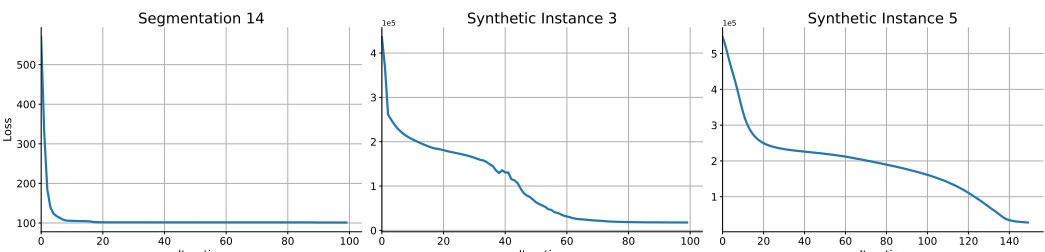

Figure 3: The loss curves of the Segmentation_14, P_potts_6 and P_potts_8 from pairwise potts synthetic problems.

## E   MORE ANALYSIS

**Efficiency vs Solution Quality.** We evaluate the performance of the NEUROLIFTING using the same network size and a consistent learning rate of 1e-4 on the Segmentation_14 dataset from the UAI 2022 inference competition, along with two of our generated Potts instances: P_potts_6 and P_potts_8. This setup allows us to observe the trends associated with changes in graph size and sparsity. The results are presented in Fig. 3. It is seen that the model converges rapidly when the graph is small and sparse, within approximately 20 iterations on the Segmentation_14 dataset. Comparing P_potts_6 and P_potts_8, we observe that though both graphs are of the same size, the denser graph raises significantly more challenges during optimization. This indicates that increased size and density can complicate the optimization process, and NEUROLIFTING would need more time to navigate a high quality solution under such cases.

## F   GNN FORMULATIONS

We summarized the popular GNN message passing formats in Table. 8 to show the logic behind the GNN backbone selection of our work.

Table 8: Graph convolutions in typical GNNs

|  | Graph Convolutions | Neighbor Influence |
|---|---|---|
| GCN | $h_i^{(k)} = \sigma\left(W_k \cdot \sum_{j \in \mathcal{N}(i) \cup \{i\}} (\deg(i)\deg(j))^{-1/2} h_j^{(k-1)}\right)$ | **Unequal** |
| GAT | $h_i^{(k)} = \sigma\left(\sum_{j \in \mathcal{N}(i) \cup \{i\}} \alpha_{i,j} W_k h_j^{(k-1)}\right)$ | **Unequal** |
| GraphSAGE | $h_i^{(k)} = \sigma\left(W_k \cdot h_i + W_k \cdot (|\mathcal{N}(i)|)^{-1} \sum_{j \in \mathcal{N}(i)} h_j^{(k-1)}\right)$ | **Equal** |

# G RESULTS ABOUT THE ANALYSIS EXPERIMENTS

In this section, we present the result figures from the analysis in Section. 5.4. These include the loss landscape visualization (Fig. 4), the average loss across different GNN backbones(Fig. 5), the comparison of different optimizers during training(Fig. 6), and the inference time comparison with Toulbar2 on different size of PCI problems(Table. 9).

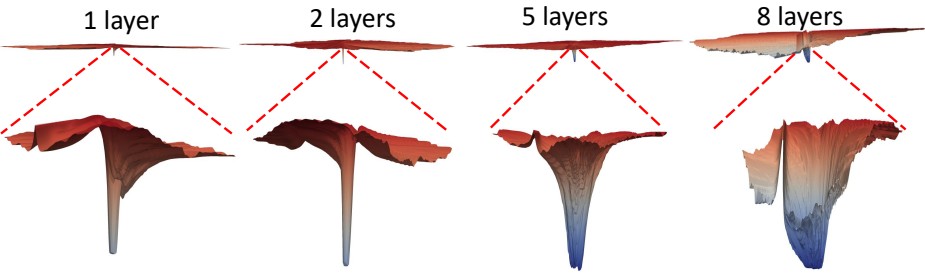

Figure 4: The landscape of instance Segmentation_19. From top to the bottom, each column correspond to network layer $\{1, 2, 5, 8\}$. The first row is the landscape range from $[-10, +10]$ for both $\delta$ and $\eta$ direction. The second row is the landscape range from $[-1, +1]$ for both $\delta$ and $\eta$ direction.

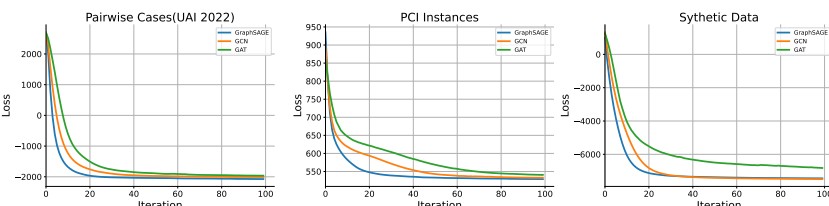

Figure 5: The average loss curves over UAI inference competition 2022 pairwise cases, PCI instances and synthetic instances using GraphSAGE, GCN and GAT as the GNN backbones.

# H VISUALIZATION SETUP

The core idea of the visualization technique proposed by Li et al. (2018) involves applying perturbations to the trained network parameters $\theta^*$ along two directional vectors, $\delta$ and $\eta$: $f(\alpha, \beta) = L(\theta^* + \alpha\delta + \beta\eta)$. By doing so, we can generate a 3-D representation of the landscape corresponding to the perturbed parameter space.

We sampled 250000 points in the $\alpha - \beta$ plane, where both $\alpha$ and $\beta$ range from -10 to 10, to obtain an overview of the loss function landscape. Subsequently, we focused on the region around the parameter $\theta^*$ by sampling an additional 10,000 points in a narrower range, with $\alpha$ and $\beta$ both from $-1$ to 1.

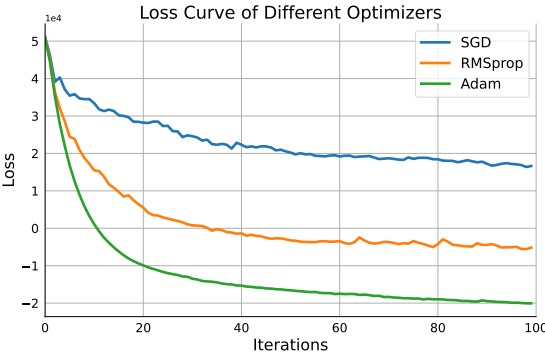

Figure 6: The average loss curves over UAI inference competition 2022 pairwise cases using different optimizers.

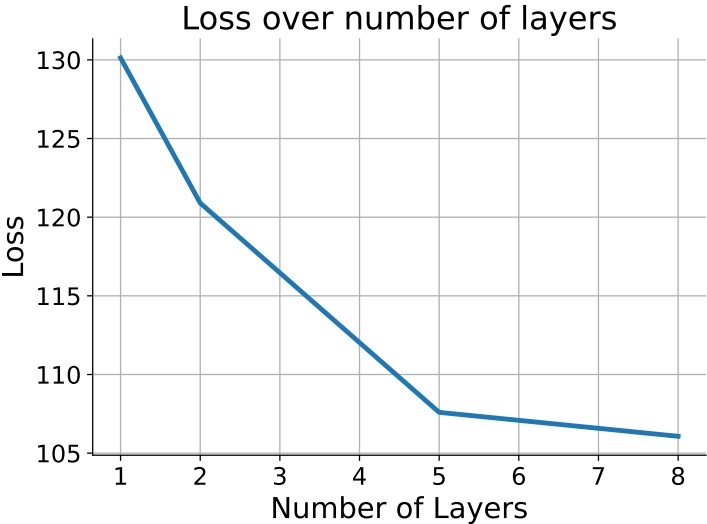

Figure 7: The training loss of instance Segmentation_19 after convergence of using network layer number $\{1, 2, 5, 8\}$.

# I    READ UAI FORMAT FILES

An example data file in UAI format is provided in Box. I. This Markov Random Field consists of 3 variables, each with 2 possible states. Detailed information can be found in the box, where we illustrate the meanings of different sections of the file. Notably, in the potential section, the distributions are not normalized. During the BP procedure, these distributions will be normalized to prevent numerical issues. However, in the energy transformation phase, we will utilize these values directly.

Table 9: Time comparison between Toulbar2 and NEUROLIFTING on PCI instances."-" if the solving process is already stopped.

| Instances | Algorithm/Solver | 200s | 400s | 600s | 800s | 1000s | 1200s |
|---|---|---|---|---|---|---|---|
| PCI_1 | Toulbar2 | 18.134 | - | - | - | - | - |
| | NEUROLIFTING | 18.211 (10s in total) | - | - | - | - | - |
| PCI_2 | Toulbar2 | 98.364 | - | - | - | - | - |
| | NEUROLIFTING | 98.446 (16s in total) | - | - | - | - | - |
| PCI_3 | Toulbar2 | 1003.640 | - | - | - | - | - |
| | NEUROLIFTING | 1003.640 (71s in total) | - | - | - | - | - |
| PCI_4 | Toulbar2 | 428.299 | 426.806 | 426.806 | 426.806 | 426.806 | 426.806 |
| | NEUROLIFTING | 408.508 | 407.6304 | 407.419 | - | - | - |
| PCI_5 | Toulbar2 | 1128.244 | 1121.325 | 1121.325 | 1121.325 | 1121.325 | 1121.325 |
| | NEUROLIFTING | 1222.281 | 1086.899 | 1077.858 | 1074.3094 | 1070.8013 | 1069.875 |

---

**Example.uai**

MARKOV      *//Instance type*
3    *//Number of vairables*
2 2 2    *//State number of each variable*
5    *//Number of cliques that has potentials*
1 0    *//1 means this clique is a variable, and the variable is 0.*
1 1
1 2
2 0 1    *//2 means this clique is an edge, the edge is (0, 1).*
3 0 1 2    *//3 means this clique includes 3 variables, and the clique is (0, 1, 2).*

2    *//The number 2 indicates that the potential in the next line has two values.*
0.1 0.9    *//The potential of variable 0 is 0.1 for state 0 and 0.9 for state 1.*

2
0.1 10

2
0.5 0.5

4
0.1 1.0 1.0 0.1*//The potential of the state combinations for variables 0 and 1 is given in the order of (0,0), (0,1), (1,0) and (1,1).*

8
0.1 2.0 0.1 0.1 0.1 0.1 0.1 2.0    *//The potential of the state combinations for variables 0, 1, and 2 is given in the order of (0,0,0), (0,0,1), (0,1,0), (0,1,1), (1,0,0), and so on.*

---

Since the transformation of variable energies and clique energies follows the same procedure, we will use the edge $(0, 1)$ to illustrate the transformation. The value calculations will adhere to Eq. 1. In Table. 10, we present the unnormalized joint distribution for the edge $(0, 1)$, while Table. 11 displays the energy table for the edge $(0, 1)$ after transformation.

Table 10: $P(x_0, x_1)$

| $x_0$ \ $x_1$ | 0 | 1 |
|---|---|---|
| 0 | 0.1 | 1.0 |
| 1 | 1.0 | 0.1 |

Table 11: $\theta_C(x_0, x_1)$

| $x_0$ \ $x_1$ | 0 | 1 |
|---|---|---|
| 0 | 2.303 | 0 |
| 1 | 0 | 2.303 |

## J  PCI PROBLEM FORMULATION

The Mixed Integer Programming format of PCI problems is as follows:

$$\min_{z,L} \quad \sum_{(i,j)\in\mathcal{E}} a_{ij} L_{ij} \tag{9}$$

$$\text{s.t.} \quad z_{np} \in \{0,1\}, \quad \forall n \in N, p \in P \tag{9a}$$

$$\sum_{p\in P} z_{np} = 1, \quad \forall n \in N. \tag{9b}$$

$$\sum_{p\in M_{ih}} z_{n_i p} + \sum_{p\in M_{jh}} z_{n_j p} - 1 \le L_{ij}, \forall (i,j) \in \mathcal{E}, \forall h \in \{0,1,2\}. \tag{9c}$$

where $n$ is the index for devices, and $N$ is the set of these indices. $P$ stands for the possible states of each device. $M_{ih}$ stands for the possible states set for node $n_i$. $L_{ij}$ is the cost when given a certain choices of the states of device $i$ and device $j$, $a_{ij}$ is the coefficient of the cost in the objective function. There is an $(i,j) \in \mathcal{E}$ means there exists interference between these two devices.

When using MRF to model PCI problems, each random variable represent the identity state of the given node and the interference between devices would be captured by the pairwise energy functions. Next we will introduce how to transform the PCI problem from MIP form to MRF form.

In the original MIP formulation of the PCI problems, three types of constraints are defined. By combining Eq. 9a and Eq. 9b, we establish that each device must select exactly one state at any given time. Furthermore, the constraint in Eq. 9c indicates that interference occurs between two devices only when they select specific states. The overall impact on the system is governed by the value of $L_{ij}$ and its corresponding coefficient. Given that interference is always present, the objective is to minimize its extent.

To transform these problems into an MRF framework, we utilize Eq. 9b to represent the nodes, where each instance of Eq. 9a corresponds to the discrete states of a specific node. The constraints set forth in Eq. 9a and Eq. 9b ensure that only one state can be selected at any given time, thus satisfying those conditions automatically. By processing Eq. 9c, we can identify the edges and their associated energies. If $z_{n_i p}$ and $z_{n_j p}$ appear in the same constraint from Eq. 9c, we can formulate an edge $(i,j)$. By selecting different values for $z_{n_i p}$ and $z_{n_j p}$, we can determine the minimum value of $L_{ij}$ that maintains the validity of the constraint.

The product of $L_{ij}$ and $a_{ij}$ represents the energy associated with the edge $(i,j)$ under the combination of the respective states. Once the states of all nodes are fixed, the values of the edge costs also become fixed. This leads to the conclusion that the objective function is the summation of the energies across all edges. Since the PCI problems do not include unary terms, we can omit them during the transformation process. This establishes a clear pathway for converting the MIP formulation into an MRF representation, allowing us to leverage MRF methods for solving the PCI problems effectively.

Table 12: $E(x_1, x_2)$

| $x_1$ \ $x_2$ | $z_{21}$ | $z_{22}$ | $z_{23}$ |
|---|---|---|---|
| $z_{11}$ | 1 | 0 | 0 |
| $z_{12}$ | 0 | 0 | 1 |
| $z_{13}$ | 0 | 1 | 0 |

Table 13: $E(x_2, x_3)$

| $x_2$ \ $x_3$ | $z_{31}$ | $z_{32}$ | $z_{33}$ |
|---|---|---|---|
| $z_{21}$ | 3 | 0 | 0 |
| $z_{22}$ | 0 | 3 | 0 |
| $z_{23}$ | 0 | 0 | 3 |

**Example**

The original problem is

$$
\begin{aligned}
\min_{z,L} \quad & L_{1,2} + 3L_{2,3} \\
\text{s.t.} \quad & z_{np} \in \{0,1\}, \qquad\qquad \forall n \in \{1,2,3\}, p \in \{1,2,3\} \\
& \sum_{p \in P} z_{np} = 1, \qquad\qquad\qquad \forall n \in \{1,2,3\}. \\
& z_{11} + z_{21} - 1 \leq L_{1,2} \\
& z_{13} + z_{22} - 1 \leq L_{1,2} \\
& z_{12} + z_{23} - 1 \leq L_{1,2} \\
& z_{21} + z_{31} - 1 \leq L_{2,3} \\
& z_{22} + z_{32} - 1 \leq L_{2,3} \\
& z_{23} + z_{33} - 1 \leq L_{2,3}
\end{aligned}
\tag{10}
$$

Then the corresponding MRF problem is

$$
\min \theta_{1,2}(x_1, x_2) + \theta_{2,3}(x_2, x_3)
\tag{11}
$$

the energy on edge $(x_1, x_2)$ and edge $(x_2, x_3)$ are shown in Table. 12 and Table. 13.

## K  LLM STATEMENT

In our research process, we utilized large language models primarily to support technical writing aspects rather than for generating research content. These tools were employed specifically for grammar checking, spell correction, improving sentence structure, and enhancing the overall readability of the manuscript. All scientific contributions, technical analyses, experimental designs, and conclusions presented in this paper are the original work of the authors, with language models serving only as writing assistance tools.

