# OpenReview forum: "NeuroLifting: Neural Inference on Markov Random Fields at Scale"
_ICLR.cc/2026/Conference — Submitted to ICLR 2026_

### Official Review · Reviewer_n3Ha · 2025-10-17

**Soundness:** 3
**Presentation:** 2
**Contribution:** 2
**Rating:** 4
**Confidence:** 3

**Summary:**

The authors introduce NeuroLifting as a method for immersing a Markov Random Field (MRF) into a high-dimensional and continuous space through a GNN. This GNN computes a probability distribution over the support of each variable within the MRF, and a mean-field approximation of the energy function is used as a surrogate (differentiable) loss function. Then, a stochastic gradient-based (SGD) algorithm (e.g., Adam) is used to search for the minimal-energy variable assignment in this MRF. Experiments on both synthetic and real-world data suggest that NeuroLifting outperforms both approximate and exact methods when dealing with large-scale MRFs.

Although interesting and seemingly effective, the paper has two main issues (detailed in the Weaknesses section). Firstly, the description of the method lacks mathematical rigor; in particular, I found Sections 4.2 and 4.3 quite difficult to follow, and the complexity analysis in Section 4.5 disregards the number of iterations for the SGD. Secondly, I believe the experiments could also benefit from a more systematic description; while multiple hyperparameter configurations have been tested (architectures, embedding dimensions, optimizer, etc.), the model’s sensibilities to these design choices remain unclear (and Table 1 lacks error bars).

**Strengths:**

1. The problem of finding the minimal-energy variable assignment in an MRF is clearly described.

2. Figures are also very helpful in understanding the proposed method.

3. Code has been released.

4. Comprehensive experiments provide a clear picture of NeuroLifting’s potential when compared to existing methods.

**Weaknesses:**

1. I found the Padding & Masking description in Section 4.2 somewhat confusing. As far as I understand, the probability of the padded values is set to the maximum energy for the corresponding variable, which is claimed to be beneficial. On the other hand, the padded values are also masked from the softmax layer. In this case, however, I think that the energy of the padded variables should not matter?

2. As I understand, the vectorizing procedure for the energy function inevitably introduces an exponential complexity to the algorithm on the size of the variables’ support. Could the authors discuss, for example, the average clique size on the experiments and how to accommodate large cliques into the proposed method?

3. On a related note, the meaning of $\psi(C_k)$ in Equation (6) is unclear; why is it not softened (like $v_i(\theta)$)?

4. As I commented above, it is wrong to claim in Section 4.5 that the algorithm has linear complexity, or that it is scalable due to the linear complexity of a SGD iteration. Indeed, the number of iterations for convergence could be (and I believe it is - due to the NP-hardness of the MRF optimization) exponentially large on the problem’s size. Besides that, $\mathcal{N}_v$ could be as large as $|\mathcal{V}|$ itself; the complexity of the iteration is, at least, $|\mathcal{V}|^{2}$ (unless further constraints are imposed).

5. Experiments are clear and comprehensive, but I believe that the work would be strengthened if not only the upper time limits were described but the actual run times for the algorithms were presented. Could the authors include (e.g., for Table 1, or any other table) the runtimes for the algorithms?

6. Also, please include error bars in Table 1.

**Questions:**

Section 4.2 states that the “high-order” graph defined by the MRF should be converted into a pairwise one. However, it is unclear whether this projection reduces NeuroLifting's expressivity. Could the authors elaborate briefly on this? Is NeuroLifting guaranteed to find a solution, given enough time and resources?

---

### Official Review · Reviewer_hhiQ · 2025-10-27

**Soundness:** 2
**Presentation:** 3
**Contribution:** 2
**Rating:** 4
**Confidence:** 3

**Summary:**

This paper proposes NEUROLIFTING, an inference method for Markov random fields (MRFs). The main idea of the method is to use graph neural networks to reparameterize MRFs, transforming the inference problem to a neural network learning problem. The proposed method is named NEUROLIFTING because its main idea is similar to the lifting method in optimization. Experiment results show that the proposed method can learn better energies.

**Strengths:**

1. The proposed method is efficient. It enables inference of MRFs with back-propagation.
2. The experiment results show that the NEUROLIFTING can learn better energies.

**Weaknesses:**

1. There is no theoretical analysis for the proposed method.
2. The experiments have some flaws.

Details can be found in the Questions section.

**Questions:**

About the theoretical analysis

1. Belief propagations have many theoretical analyses, e.g., convexity and convergence. Could the authors provide some analysis of the proposed method?

About experiments.

1. Are there better ways to show the effectiveness of the proposed method? That is, it is hard to tell how well the proposed method performs based on energy values alone. Are there any methods to show improvements more intuitively, e.g., node classification accuracy?

2. BP can also run on GPUs [1], and is also fast. Could the authors also compare with it?

3. I am not familiar with the Toulabr2. Why can the proposed method learn better energy values than the exact inference method? Isn’t it that the exact inference method can learn the optimal solution? This is a little counterintuitive.

[1] Reid Bixler et al. Sparse-Matrix Belief Propagation. UAI 2018.

---

### Official Review · Reviewer_QhTi · 2025-11-05

**Soundness:** 3
**Presentation:** 1
**Contribution:** 2
**Rating:** 2
**Confidence:** 3

**Summary:**

The paper presents a new method for MAP inference in Markov random fields, utilizing Graph Neural Networks (GNNs) to parametrize a lifter version of the optimization problem. A procedure is outlined for how to frame the energy minimization as a GNN training process, with learnable node embeddings and a soft version of the energy function as loss. Experiments are carried out on synthetic MRFs, examples from the UAI competition and an example 5G network cell problem. The method overall performs competitive with the baselines.

**Strengths:**

1. The idea of applying lifting to MAP inference in MRFs is interesting, and the use of GNNs to parametrize the solution natural and well-motivated.
2. The set of experiments considered is extensive, and includes a large set of different MRFs. Relevant baselines are considered.
3. The authors make a good motivation for why Graph-SAGE is a good choice of GNN for this specific task, and also support this with experimental evaluation.
4. For many experiments the method finds good solutions, and beating even the strong Toulbar2 baseline.

**Weaknesses:**

1. The biggest weakness of the paper is an overall unclear presentation, using too much jargon and not focusing on the key points of the method. In particular:
1.1 There are multiple non-standard abbreviations that are never explained, or explained after they have already been used before.
1.2 It took me until the experiment section until I understood if the idea is to train one GNN that can perform (amortized) inference for multiple MRFs, or train one new GNN per MRF (this seems to be the case).
1.3 It is hard to get an overview of the full method, despite a long description in section 4. Figure 1 is helpful, but could be substantially flashed out to explain the many steps involved. There is also discussion of quite obvious pre-processing steps (e.g. one-hot-encoding), rather than focusing on the core part of the method. Things like "jumping knowledge" are simply mentioned without any description, making it completely unclear how central this is to the method. I would advice to make it clear 1) in exact order what are the steps in the method, 2) what are the inputs and outputs of the GNN. Both these things are now explained a different places in the text, but never coherently and concisely.
1.4 It is also hard to get an overview of the results, as much of it is completely deferred to appendix. It would also be better to present some form of summary results, rather than per-instance energy values for each MRF. The tables become quite large and challenging to interpret.
2. The abstract and intro presents the method as for inference in MRFs, but when getting to section 3 it is clear that the problem being tackled is specifically MAP estimation in MRF with discrete-valued random variables. This is a lot more specific and not the only type of inference relevant for MRFs. The introductory parts should reflect this.
3. Around line 185 the authors write "To facilitate the power of GNNs, we need to convert high-order graph into a pairwise one.". There is a large literature on higher-order GNNs for hypergraphs, which seems to directly solve this issue and would potentially be more suitable here. See for example [1] for an overview. While one can argue that this can be left for future work, this literature should at least be acknowledged.
4. The method overall shows good performance, often comparable to the baselines. However, the presentation makes it seem like the key benefit is that the proposed method is more efficient. Still, while the computational time of Toulbar2 is extensively discussed, I find no runtimes reported for the actual proposed method. This is central to the relevance of the method, given the results. There is a theoretical complexity analysis given, but this only discusses the computational complexity of a single training iteration. It misses the crucial point that the number of steps we need to train for (optimizer iterations) will be crucial for the efficiency of the method. Even with the number of iterations explicitly given, I have no way to compare this to the other methods as their runtime is given in seconds rather than GNN training iterations.

Minor issues:
1. Something seems wrong with the math typesetting, with equations showing up in too small font and weirdly aligned. See in particular eq. 2. Some passages, in particular in the experiments, could use more references. In particular pointing to where datasets come from.
3. The paper could make use of an additional proofreading and grammar check. There are not many such problems, but they do stand out when reading through it.
4. The text in figure 5 is too small to read.

[1] Kim, Sunwoo, et al. "A survey on hypergraph neural networks: an in-depth and step-by-step guide." Proceedings of the 30th ACM SIGKDD Conference on Knowledge Discovery and Data Mining. 2024.

**Questions:**

None

---

### Meta-Review · Area_Chair_Q7CS · 2026-01-05

**Summary:**

The reviewers generally agree that, although the method achieves competitive empirical performance, the paper suffers from unclear presentation and insufficient methodological clarity. The overall pipeline, problem scope, and key design choices are difficult to follow, with heavy use of unexplained terminology and an imprecise framing of the task (general inference vs. MAP estimation). In addition, claims about efficiency and scalability are not well supported, as theoretical analysis is limited and actual runtime comparisons are missing. While the experiments are extensive, their presentation lacks clarity, with limited summary statistics and missing error bars, which weakens the overall evaluation.Moreover, the authors did not provide any response during the rebuttal stage. Considering these issues, I recommend rejection.

**Reviewer Concerns:**

The authors did not provide any reply in the rebuttal.

**Reviewer Scores:**

N/A

---

### Decision · Program_Chairs · 2026-01-26

Reject